# Statistical Modeling and Optimization of Electrospinning for Improved Morphology and Enhanced β-Phase in Polyvinylidene Fluoride Nanofibers

**DOI:** 10.3390/polym15224344

**Published:** 2023-11-07

**Authors:** Asra Tariq, Amir H. Behravesh, Ghaus Rizvi

**Affiliations:** Faculty of Engineering and Applied Science, University of Ontario Institute of Technology, Oshawa, ON L1G 0C5, Canada

**Keywords:** electrospinning, Taguchi design method, piezoelectric, PVDF, β-phase

## Abstract

The fabrication of PVDF-based nanofiber mats with enhanced β-phase using electrospinning and post processing was optimized using Taguchi design methodology. The parameters studied include the concentration of PVDF in the DMF (Dimethylformamide) solvent, applied voltage, flow rate, and drum speed. A reliable statistical model was obtained for the fabrication of bead-free PVDF nanofibers with a high fraction of β-phase (F(β)%). The validity of this model was verified through comprehensive regression analysis. The optimized electrospinning parameters were determined to be a 23 wt% PVDF solution, 20 kV voltage, a flow rate of 1 mL/h, and a drum speed of 1200 revolutions per minute.

## 1. Introduction

Electrospinning is a highly versatile additive manufacturing process capable of fabricating a wide variety of materials into nanoscale fibers, with and without fillers [1]. Nanofibers produced by electrospinning are gaining prominence as an emerging method for processing materials for numerous applications, owing to the inherent advantages of their low weight, high surface area, exceptional flexibility, conformability, and easy integration into wearable/soft devices [2]. These advantages are resulting in increased use of electroactive polymeric fibers for developing sensors and actuators.

Polyvinylidene fluoride (PVDF) and its copolymers are highly flexible and good piezoelectric materials recommended by researchers for sensors, actuators, and energy-harvesting devices. PVDF possesses a notably low weight, and it is easy to process because of its thermoplastic nature. Its flexibility allows it to readily adapt to the contours of the wrist or curved body parts, facilitating biomechanical analysis [3]. The crystalline structure of PVDF exhibits five different polymorphs, i.e., α, β, γ, δ, and ε phases, based on the conditions of crystallization [4]. The β-phase in PVDF exhibits a greater piezoelectric effect, and hence fibers rich in this phase are desirable for making sensors and actuators [5,6,7]. The β-phase is generated by the arrangement of fluorine (F) atoms on one side of the carbon chain and hydrogen (H) on the opposite side. This phase has the highest dipole moment per unit cell and offers an excellent piezoelectric coefficient (d33 = 49.6 pm/V). A high piezoelectric coefficient offers greater sensitivity to external mechanical stimuli [8].

The majority of prior research focused on PVDF’s application in energy harvesting and sensors has predominantly relied on PVDF films prepared through the film-casting method. Tanaka et al. developed a PVDF film-based vibrational sensor for measuring skin vibrations. This prepared film demonstrated the ability to discern between different levels of surface roughness as the PVDF sensor was moved across it with a finger [9]. Seminara et al. fabricated arrays of piezoelectric polymer transducers employed in artificial skin, operating within a frequency range spanning from 1 Hz to 1 kHz [10]. It was observed that the film-casting method offers a major α-phase of PVDF and less than 40% β-phase fraction. A decrease in the size scale of PVDF results in a dramatic increase in the sensing ability [11]. Electrospun PVDF nanofibers present a higher surface area in comparison to amorphous thin films and offer higher sensitivity to mechanical stimulus [12,13]. During the electrospinning process, PVDF chains undergo stretching, facilitated by a strong electric field, which promotes a greater concentration of the β-phase within the fibers [14,15,16]. The electrospinning of PVDF was carried out by many researchers and the effect of the electrospinning process’ parameters on the morphology of nanofibers (fiber diameters, porosity, and uniformity of fibers) was explored [17,18,19,20,21,22]. Sita M Damaraj et al. prepared a nanofiber mat of PVDF based scaffolds by varying the voltage in a range of 12–30 kV and keeping the remaining process parameters constant (20 wt% PVDF in 1:1 solvent mixture of DMAC and acetone, 0.5 mL/h flow rate, flat-plate collector at a distance of 20 cm from the needle). The increase in the voltage reduced the fiber diameter from 295 nm to 151 nm and increased β-phase from 65% to 72% [19]. On the contrary, C. Ribeiro et al. discovered that increasing the voltage between 14–30 kV for 20 wt.% PVDF in DMF/Acetone solvent mixture at 4 mL/h flow rate resulted in the reduction in β-phase from 85–80% [6]. Hao Shao et al. investigated the impact of varying PVDF concentration in a DMF/Acetone mixture (with a ratio of 4/6 *v*/*v*) within the range of 16–26 wt.% while maintaining a flow rate of 1 mL/h. 15 kV applied voltage and a drum speed of 100 rpm. The study revealed that as the concentration increased, there was a corresponding rise in fiber diameter within the range of 0.2–0.8 μm. The β-phase content initially increased upon raising the concentration to 20 wt.% but subsequently began to decrease as the polymer concentration was further elevated [23]. The effect of solvent mixture on the piezoelectric property of PVDF was studied by S. Gee et al. Based on the findings, it was deduced that electrospinning a 12 wt.% PVDF solution at 9 kV voltage and a flow rate of 0.75 mL/h in a mixture of DMF/Acetone with a 60/40 ratio resulted in an impressive 89% β-phase fraction (%F(β)). However, the nanofibers were coarse (400–600 nm) with very high irregularity (S.D = 300 nm) [24]. Bilal Zaarour et al. described that when PVDF is electrospun using acetone or a DMF and acetone mixture it exhibits reduced piezoelectric sensitivity and consequently generates lower voltage compared to when pure DMF is used, despite having a higher β-phase content. This is attributed to the small fiber diameter and bead-free fibers achieved when electrospinning with DMF alone [25]. High-beaded nanofibers negatively impacts the mechanical properties and reduces the β-phase content in PVDF, as it prevents adequate fiber stretching [26]. R.K. Singh et al. explained the effect of needle size and flow rate of solution on the β-phase of PVDF. The inner diameter of a needle higher than 0.36 mm resulted in the reduction in β-phase from 75% to 65%. Flow rate higher than 2 mL/h for 20 wt% of PVDF (534,000 molecular weight) caused a drop in β-phase [12]. The type of collector also affects the structure and alignment of nanofiber during electrospinning, so it also affects the β-phase of PVDF. It was found that a rotating collector offers high β-phase compared to a static plate-type collector because of high stretching in fiber when collected [17].

All the prior research was conducted across a wide range of process parameters, resulting in a significant variation in the produced fiber size, and researchers have reported diverse outcomes. This occurs because altering a single parameter in the electrospinning process can have a profound impact. Attaining desirable fiber morphology and high β-phase requires the careful alignment of all parameters. In this study, this is achieved by establishing some fixed process parameters that consistently yield good morphology and F(β) in the literature, while systematically varying other parameters where there is divergence of opinion. Within a narrow range of these process parameters, minimal disruption occurs, resulting in the production of fibers with minimum fiber diameter and a high β-phase content. The key objective of this study is to identify the optimal combination of acceptable parameter ranges within the electrospinning process for PVDF. The aim is to maximize key response variables such as β-phase content, reduced fiber diameter, and improved fiber uniformity. A statistical model is developed that establishes a correlation between these process parameters and the critical response variables, specifically β-phase fraction, and fiber diameter. These variables are of paramount importance, particularly in the context of PVDF’s piezoelectric properties, especially for sensor applications.

The selection of solvent, PVDF molecular weight, electrospinning needle size, and collector type was made after a comprehensive review of the literature, where no significant variation in outcomes was observed for these factors. Needle tip-to-collector drum distance (TCD) was 13 cm and was constant for all samples.

## 2. Materials and Sample Preparation

Polyvinylidene fluoride (PVDF)—Kynar 740 molecular weight: 280,000 g/mol from Arkema group, Burlington, ON, Canada. N, N dimethylformamide (DMF) with 99.8% purity was obtained from ACP Chemicals, Saint-Léonard, QC, Canada. PVDF solution was prepared by following the process steps described in Figure 1.

Figure 2 illustrates the laboratory-scale electrospinning setup constructed for this study, comprising the following essential components: (a) high-voltage power supply (Gamma High Voltage Research Inc.—Ormond Beach, FL, USA), (b) syringe pump (New Era Company—Buffalo, NY, USA), (c) rotating cylindrical collector drum with adjustable speed control, and (d) needle holder with forward and backward motion capability. The PVDF + DMF solution was loaded into a 5 mL syringe and a 25 G needle (outer diameter = 0.5 mm, inner diameter = 0.26 mm) was securely attached to the positive terminal of the voltage power supply. The flow rate of the polymer solution was controlled using the syringe pump, while the voltage was carefully regulated by the power supply. For safety precautions, the entire assembly was enclosed within an acrylic box to shield against high voltage exposure and exposure to solution vapors. Each sample was electrospun for 3 h and the thickness of the prepared mat (calculated by VHX-1000 Digital Microscope—Keyence, Osaka, Japan) was between 150–200 μm. 

The design of experiments (DOE) for modeling and optimizing the electrospinning process parameters of PVDF nanofibers was executed using Taguchi’s orthogonal array (OA) design. The residual analysis in the regression model was performed to predict the values of F(β)%. Taguchi’s OA design is simple and robust and was used in the study for fiber diameter and β-phase optimization. This study investigates the effect of four electrospinning parameters: (a) high voltage, (b) polymer concentration, (c) speed of collector drum, and (d) solution flow rate (shown in Table 1). Each parameter was subjected to three levels of variation.

The design consists of 9 rows that require experimental trials shown in Table 2. The mean output performance and its variation were assessed using Taguchi’s design analysis.

### Characterization

The study employed various characterization techniques, including the use of a Hitachi FlexSEM 1000 (by High Technologies America, Clarksburg, MD, USA) scanning electron microscope (SEM) to analyze the morphology of electrospun PVDF nanofibers. Perkin Elmer Fourier transform infrared spectroscopy (FTIR) and a Bruker X-ray diffractometer were employed to assess and quantify the relative fraction of β-phase (%F(β)) in the PVDF nanofibers. A Discovery HR20 stress-controlled rheometer produced by TA Instruments, New Castle, DE, USA, was used for measuring the viscosity of PVDF solutions. Desktop Conductivity Meter by Mettler Toledo (located in Greifensee, Switzerland) was used for measuring the electrical conductivity of PVDF solutions.

## 3. Results and Discussion

### 3.1. Morphology of PVDF Nanofibers

Figure 3a presents the SEM image of nanofibers corresponding to P-6 (with electrospinning conditions at 20 kV, 23 wt.%, 1200 rpm, and 1 mL/h). Figure 3b represents a magnified segment derived from the SEM image shown in Figure 3a. The SEM images were processed and analyzed using the ImageJ tool, with the resulting fiber diameter of P-6 depicted in Figure 3b. Fiber diameter was measured for a total of 100 fibers at different locations of mat and a histogram was plotted to analyze the uniformity of fiber structure. For some electrospinning parameters, substantial bead formation occurred, with a broad fiber diameter range.

In electrospinning, lower fiber diameter principally leads to an increase in specific surface area. An increase in the surface area of fiber offers excellent mechanical properties, such as tensile strength and flexibility. A decrease in the fiber diameter of PVDF nanofibers can amplify the voltage generated under a specific force, thereby elevating the material’s sensitivity [11,28].

Nanofibers can never be in an ideal regular form without any defects, therefore measuring the average fiber diameter is not sufficient. Distribution of the nanofibers is required to analyze while selecting the right parameters of electrospinning. Figure 4 compares the fiber uniformity of samples P-6 and P-8. It was observed that an increase in the polymer concentration resulted in a more regular fiber diameter without bead formation. The presence of a beaded structure can occur as a result of polymer solution instability during spinning. This instability arises from surface tension, which compels a liquid to adopt a reduced surface area per unit mass, typically taking on a spherical shape. Higher viscosity promotes the formation of bead-free fibers [29]. Samples P-3 and P-6, both prepared with 23 wt% PVDF and voltage settings of 17 kV and 20 kV, respectively, exhibited uniformity in fiber diameter. In contrast, when the PVDF solution concentration was lower, as in the case of P-8, a higher irregularity was observed due to the instability of the Taylor cone [30].

Figure 5 shows the SEM images of P-3 (23 wt.%, 17 kV, 1.4 mL/h at 1700 rpm) and P-9 (23 wt.%, 23 kV, 1.2 mL/h at 700 rpm). It can be demonstrated that an increase in voltage results in increased irregularity, even at a high polymer concentration. Unevenness in fiber diameter will result in weak links between fibers inside the mat which remarkably raises the possibility of lower mechanical properties. At high voltages, the Taylor cone can become unstable. This instability can lead to fluctuations in the flow of the solution and cause variations in fiber diameter and morphology [22].

Figure 6 demonstrates the effect of electrospinning parameters on the average fiber diameter analyzed by Taguchi. It was observed that the effect of solution concentration on fiber diameter was ranked first (additional information is provided in Appendix A). An increase in the PVDF concentration resulted in thicker fiber. The cohesion forces between polymer chains started to increase and gave resistance to the stretching of fiber before solvent evaporation [31]. In conventional solution electrospinning, it is a well-established fact that enhancing the viscosity of the solution tends to reduce the occurrence of bead formation [32,33]. The viscosity of all the blends shown in Figure 7 was measured from 10 to 1000 s^−1^ at 22 °C using a 40 mm plate–plate geometry. The viscosity of the solution increases by increasing the concentration of PVDF in DMF. Traditionally, polymers behave such that on increase of the shear rate the viscosity starts to decrease due to the shear thinning behavior of polymers. In high-viscosity polymer solutions, the polymer chains are typically disordered. When shear stress is applied, the chains tend to align in the direction of the shear. As a result, the solution’s resistance to flow decreases, causing the apparent viscosity to drop. This alignment of polymer chains reduces their hindrance to flow and makes the solution less viscous under shear. In dilute solutions, the polymer concentration is low and other solvent interactions may dominate the viscosity behavior [34].

Since the exact shear rate applied to the solution during electrospinning is unknown, an approach to narrow down the range of shear rates was taken. To achieve this, the impact of polymer concentration on viscosity was analyzed at three specific shear rates: 10 s^−1^, 200 s^−1^, and 400 s^−1^ (shown in Figure 7b. The viscosity of the solution at 17 wt.% is approximately 0.5 Pa.s at all three-shear rates which is considered very low for producing bead-free fibers in electrospinning. Raising the viscosity of the solutions reduced bead formation; however, it concurrently led to an increase in fiber diameter. The reason is high viscosity led to increased viscoelastic forces that opposed axial stretching during the electrospinning process, ultimately resulting in larger fiber diameters [35].

Table 3 provides data on the electrical conductivities of PVDF solutions at various concentrations. As the solution concentration increased, the conductivity of the solutions decreased. Nonetheless, all these conductivity values remained sufficiently high for electrospinning [21].

The factor that ranked second in its influence on fiber diameter is voltage. Elevating the voltage from 17 kV to 23 kV increased fiber diameter. It is worth noting that in the literature, there are contradictory reports regarding the impact of high voltage on fiber diameter. For a fixed solution concentration, increasing the voltage from a specific value started to accelerate the electrospinning jet and draw a greater volume of solution from the tip of the needle. This eventually caused the Taylor cone to recede into the needle and generate irregular fibers and beads [36,37]. It has been observed in the literature that increasing up to a specific level of voltage results in decreased fiber diameter. This trend started to reverse after a critical value of voltage. For a specific limit of voltage, the fiber diameter will continue increasing because of reduced flight time that does not allow the fiber to stretch during the evaporation of the solution. An increase in the voltage value above that limit will again start to reduce the fiber diameter because of the division of a single jet into multiple jets. The effect of voltage on fiber morphology can be understood from Figure 8.

The influence of flow rate on fiber diameter is ranked third; the existing literature also suggests that the effect is minimal within a certain range of solution flow rates. It was observed that an increase in the flow rate led to thicker fibers and an increased occurrence of bead formation. This is due to the increased drop volume and initial radius of the electrospinning jet consequently leading to an increase in fiber diameter [39,40,41]. The least effect noted was the speed of the collector drum varied from 700 rpm to 1700 rpm. An increase in the speed of the collector drum resulted in reduced fiber diameter due to high mechanical stretch causing the elongation of fiber during the deposition of the mat [42].

### 3.2. β-Phase Fraction

FTIR is primarily employed for identifying the chemical bonds responsible for vibrational motions in the molecular chains of PVDF with distinct polymorphs. Moreover, FTIR spectra are also used for quantifying the amount of β-phase content. A comparison of three different electrospun mats with different F(β)% is shown by FTIR spectra in Figure 9. Various peaks representing different chemical bonds in PVDF are highlighted, such as the vibrational bands of α-phase at 760 cm^−1^ (CF_2_ bending and skeletal bending) and 976 cm^−1^ (CH out-of-plane deformation). The bands due to β-phase are 840 cm^−1^ (CH_2_ rocking) and 1280 cm^−1^ (CF out-of-plane deformation). The rest of the peaks are explained in Table 4 [43].

The effect of electrospinning parameters on the β-phase content of PVDF was calculated by FTIR spectra using the equation [44]:$$F\left(\beta \right)=\frac{X_{\beta }}{X_{\alpha }+X_{\beta }}=\frac{A_{\beta }}{{(K_{\beta }/K_{\alpha })A}_{\alpha }+A_{\beta }}=\frac{A_{\beta }}{{1.3A}_{\alpha }+A_{\beta }}$$
where,

*A_α_* = absorbances at 766, 976, and 1240/cm (due to the α-phase of PVDF);

*A*_β_ = absorbances at 840 and 1274/cm, (due to the β-phase of PVDF);

*K*_β_ and *K_α_* are the absorbance coefficients of 7.73 × 10^4^ and 6.13 × 10^4^ cm^2^/mol, respectively.

Of these, P-3 (23 wt.%, 17 kV, 1.4 mL/h at 1700 rpm) has the highest F(β)% which is 70%, P-1 (17 wt.%, 17 kV, 1 mL/h at 700 rpm) has 67%, and P-7 (23 wt.%, 17 kV, 1.4 mL/h at 1200 rpm) showed 64%. It can be noticed that P-3 exhibited a more intense peak at 840 cm^−1^ and a very weak peak at 760 cm^−1^.

For validation, XRD analysis was also performed to identify and calculate F(β)%. Figure 10 exhibits the comparison of electrospun mat having high (P-6) and low (P-8) F(β)%. In X-ray generation, a cobalt source was employed, providing high-quality diffractograms and ensuring clear phase identification. In most of the literature, XRD analysis of PVDF was conducted using a copper source. The conversion of 2θ values was performed using TOPAS software (version 6) to facilitate result comparison with the existing literature. The experimental XRD curves were deconvoluted into individual component peaks through peak fit analysis executed in Origin Pro 2018. Figure 11 shows two distinct graphs, one representing the β-phase (P-6) and the other illustrating the coexistence of both the β-phase and γ-phase in P-8. The XRD pattern of α-PVDF exhibits the presence of diffraction peaks at 21.35°, and 31.07° relating to the (020), and (021) reflections. The XRD pattern of β-PVDF shows a characteristic peak at 23.9°, 42.19°, and 48.14° corresponding to the diffracting planes of (110)/(200), (101), and (111). The γ-PVDF exhibits characteristic peaks at 23.35° assigned to the (110) reflections, respectively. P-6 having high β-phase displayed a very intense peak of β-phase along with α-phase pattern with no prominent shoulder of γ-phase. P-8 with less measured F(β)% showed a γ-phase peak and the β-phase peak was also moved towards a slightly lower 2θ value.

Figure 12 exhibits the Taguchi analysis of the effect of electrospinning parameters on the β-phase content in a PVDF nanofiber mat (additional information is provided in Appendix A). The effect of concentration was ranked as first and the voltage in the level of 17 kV to 23 kV was ranked as the least factor affecting β-phase content in PVDF. At high concentrations, the nanofibers obtained had fewer beads and were more uniform.

At lower concentrations, such as 17 wt.%, the β-phase content was reduced due to an excess of beads, which hindered effective fiber stretching. However, an increase in the polymer amount improved the fiber’s stretchability and subsequently increased the β-phase content. The literature indicates that at exceedingly high polymer concentrations, the β-phase content begins to decrease due to reduced chain stretchability caused by the elevated entanglement among polymer chains [23].

It can be observed that the increase in the flow rate from 1 to 1.4 mL/h, resulted in the reduction in the β-phase percentage. At a high flow rate, the evaporation of solvent from fiber becomes difficult. The minimized fiber stretching leads to less β-phase content [45].

Drum speed varied from 700 rpm to 1700 rpm. It was observed that 1200 rpm is the critical value of drum speed on which the mechanical stretch on the nanofiber was enough to enhance the β-phase. Drum speed not only increases the stretching in nanofibers but also helps in the orientation of nanofibers. Higher drum speed results in a high orientation of nanofibers. It can be explained as, at very low drum collector speeds such as lower and equal to 700 rpm, there was less effective mechanical stretching that resulted in less conversion of α-phase to β-phase. At the middle level of the drum speed, which is 1200 rpm in this case, the drum speed provided optimum mechanical stretching for the conversion into the β-phase. At very high drum speed (1700 rpm), there were huge vibrations in the system that were noted during the experimentation. This speed caused air turbulence on the collector surface. This combination of high-speed rotation and air turbulence caused the fibers to be deposited without enough mechanical stretching. The rapid rotation of the drum impacts the mechanical stretching force exerted by the collector. Consequently, the deposited fibers lacked the necessary elongation to facilitate the reorientation of polymer chains from α-phase to β-phase in PVDF [45,46].

The increase in the voltage was continuously increasing the %β-phase due to high stretching on the fibers. However, this increase within the tested range (17–23 kV) was not very prominent. Overall, the maximum β-phase percentage improved by optimizing the electrospinning parameters was 75% for 23 wt% of PVDF, at 20 kV, 1 mL/h, and 1200 rpm.

The signal-to-noise ratio (SNR) serves as a key performance parameter in the Taguchi method, derived from a series of optimization experiments. It operates under two conditions: “smaller-the-better” and “larger-the-better”. In this work, electrospinning factors are varied to obtain high β-phase, so a larger-than-better SNR condition is used. This is defined as [47]:$$SNR=-10\times {log}_{10}\left(\frac{1}{n}\sum_{i=1}^{n}\frac{1}{y_{i}^{2}}\right)$$
where *y_i_* is the value of β-phase acquired from respective experiments and *n* is the number of experiments.

The larger the SNR value the higher will be the impact of a parameter on a response. The mean value of SNR varying by different levels of electrospinning factors is illustrated in Figure 13. The analysis reveals that electrospinning a solution containing 23 wt.% of PVDF in DMF solvent, with a flow rate of 1 mL/h, 23 kV voltage, and a drum speed of 1200 rpm, results in the highest β-phase content. Notably, the voltage has a relatively minimal impact in comparison. Hence, it is advisable to electrospun at 20 kV to achieve both uniform fiber diameter and a high β-phase content.

Table 5 shows the comparison of the fiber diameter as well as the β-phase reported by many researchers and the data obtained in this study.

### 3.3. Model Fitting and Validation

The mathematical model was developed for the response (β-phase content) using fit regression analysis to predict the response values. The model was developed using Minitab software (Version 21.1.0). The regression equation for predicting and defining the relationship of process variables on output response is demonstrated here:Y=ε+0.243∗Voltage+1.554∗concentration−0.00153∗Drum speed−11.98∗flow rate

Y= response (β-phase content)

ε = 48.58 is the error normally distributed on output response Y.

The graphical illustration shown in Figure 14 illustrates the comparison between the mathematical model and the experimental results, enabling an assessment of the model’s accuracy.

The null hypothesis posited that none of these tested factors had any effect on the response variable, β-phase. So, if the null hypothesis is true, it means the independent variables (electrospinning factors) in the model have no effect on the dependent variable (β-phase) and the model has no statistical significance. The adequacy of this model was validated by coefficients of multiple determination (R^2^), F-value, and P-value of the regression model. R² or the coefficient of determination of this model was observed at 91.88% for the full experimental runs at a 95% confidence interval. Which explains that 91.88% variability of independent factors is explained by the model. The F-value of the model is 11.32 which indicates that the model holds importance. Model terms with *p*-values below 0.05 are deemed significant. In this case, the model was highly accurate for concentration where the *p*-value = 0.04 and flow rate with *p*-value= 0.03. However, it poorly fit for voltage and drum speed (presented a value greater than 0.05). Therefore, as per the explanation provided above, the null hypothesis was not upheld. Overall, the calculated average accuracy falls within an acceptable range for model verification. To enhance model accuracy, one can consider either increasing the number of experimental runs or incorporating the interaction effect model.

The piezoelectric property of PVDF is highly dependent on F(β)% that can be optimized by electrospinning process parameters. The %F(β) as a function of concentration and voltage is presented in Figure 15. The results are plotted by holding drum speed and flow rate value constant at 1200 rpm and 1.2 mL/h The results depict that the %F(β) increases by increasing the concentration and voltage, however, the influence of concentration is more prominent as compared to voltage. It can be found that even at 23 kV voltage, 17 wt.% of PVDF solution does not offer more than 65% F(β). At high concentrations of PVDF in DMF solvent, fibers are more stretched. At 23 wt.% of PVDF, the solvent amount is enough to provide adequate evaporation and solidification of fibers. This intermediate evaporation of solvent causes the formation of β-phase crystals to be more favorable as compared to α or γ phases. It was also proven by researchers that the formation of high β-phase requires an intermediate speed of evaporation of the solvent. At very slow α-phase is thermodynamically and at very high evaporation, α-phase is kinetically more favorable [50]. When a high voltage is applied during the electrospinning of the PVDF solution, it exerts an electrostatic force on the solution droplet, causing the polymer chains to align along the direction of the electric field. This alignment can promote the formation of the β-phase, as it is favored by a specific chain orientation [17,18].

The %F(β) as a function of flow rate and voltage is exhibited in Figure 16. The values on hold are concentration at 20 wt.% and drum speed at 1200 rpm. The findings suggest that elevated voltage and reduced flow rate contribute to a higher β-phase content. Specifically, a combination of 1 mL/h flow rate and voltage ranging from 20–23 kV yields the maximum β-phase content. However, when the flow rate is increased to 1.4 mL/h, the β-phase content falls below 65%. The combination of a low flow rate and high voltage results in better chain orientation and crystalline alignment in the fibers, increasing the β-phase content. The high voltage and slow flow rate lead to increased stretching and strain on the polymer chains. This strain can promote strain-induced crystallization, which is known to favor the β-phase.

The influence of concentration and drum speed on %F(β), while maintaining a constant voltage of 20 kV and a flow rate of 1.2 mL/h, is depicted in Figure 17. It can be observed that very high drum speed causes a reduction in β-phase due to not enough stretch in nanofibers as a result a lot of vibration and disturbance. A speed of 1200 rpm at a high PVDF concentration resulted in the highest β-phase fraction. It is important to highlight that a PVDF concentration of 23 wt.% in DMF consistently yields a significantly higher β-phase fraction across all drum speeds, in contrast to solutions with lower wt.%.

Figure 18 illustrates the main parameters influencing β-phase content, namely the concentration and flow rate of the polymer solution while keeping voltage constant at 20 kV and drum speed at 1200. As previously discussed, and as evident in Figure 15, increasing the concentration while decreasing the flow rate has the potential to enhance the β-phase fraction.

## 4. Conclusions

In conclusion, it was observed that achieving both desirable fiber morphology and an elevated β-phase content is contingent upon carefully balancing the electrospinning parameters. The concentration of the PVDF solution played a pivotal role, with higher concentrations leading to enhanced fiber stretchability and increased β-phase content. However, excessively low concentrations resulted in bead formation and less stable fiber stretching. At a PVDF concentration of 23 wt.%, the average fiber diameter is relatively high, yet it results in more uniform and bead-free fibers. The optimal electrospinning conditions for producing smooth, bead-free nanofibers with a high β-phase were found to be a flow rate of 1 mL/h and a drum speed of 1200 rpm. Although a voltage of 23 kV resulted in the highest β-phase in PVDF nanofibers, it also led to irregular fiber diameters, potentially compromising mechanical strength. Hence, a voltage of 20 kV is recommended, which maintains a high β-phase content while ensuring uniform fiber diameters when using a 23 wt.% concentration, 1 mL/h flow rate, and 1200 rpm drum speed. Overall, the optimization of electrospinning parameters for PVDF nanofibers is a complex interplay, and careful consideration of these parameters is essential to achieve the desired combination of morphology and β-phase content. These research findings not only contribute to a better understanding of PVDF nanofiber production but also provide valuable insights for applications requiring high-quality PVDF nanofibers, such as sensors and energy harvesting devices.

## Figures and Tables

**Figure 1 polymers-15-04344-f001:**
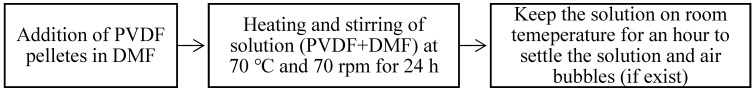
Step by Step process for making the solution of PVDF for electrospinning.

**Figure 2 polymers-15-04344-f002:**
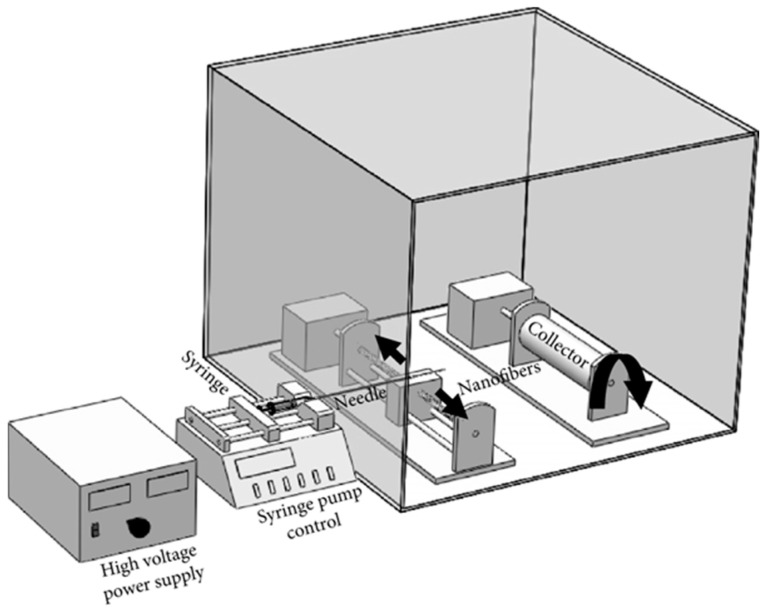
An electrospinning setup is used for preparing samples [27].

**Figure 3 polymers-15-04344-f003:**
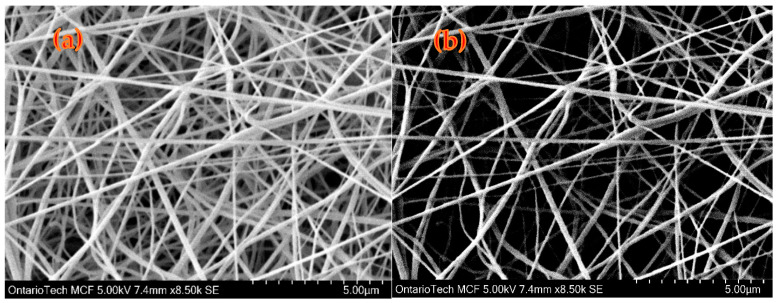
SEM images of electrospun sample (**a**) P-6 (at 20 kV, 23 wt.% PVDF, 1200 rpm, 1 mL/h). (**b**) magnified section of P-6 processed in ImageJ tool.

**Figure 4 polymers-15-04344-f004:**
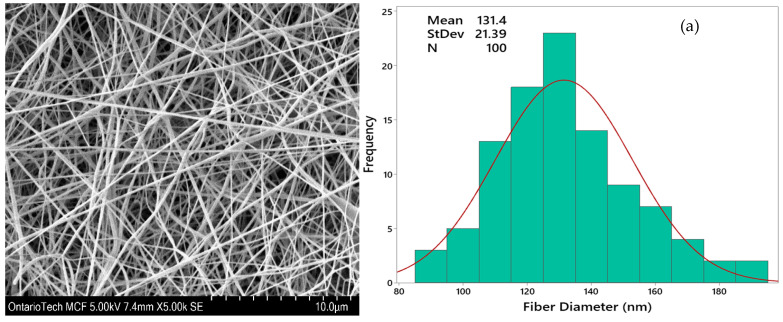
Scanning electron microscopy images and fiber diameter distribution of PVDF mat: (**a**) P-6 processed at 20 kV voltage, 23 wt.% concentration, and 1 mL/h flowrate at drum speed of 1200 rpm and (**b**) P-8 processed at 23 kV voltage, 20 wt.% concentration, and 1 mL/h flowrate at a drum speed of 1700 rpm.

**Figure 5 polymers-15-04344-f005:**
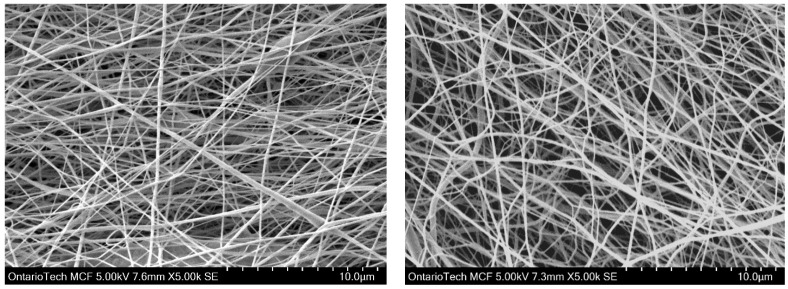
SEM image of P-3 (23 wt.%, 17 kV, 1.4 mL/h at 1700 rpm) and P-9 (23 wt.%, 23 kV, 1.2 mL/h at 700 rpm).

**Figure 6 polymers-15-04344-f006:**
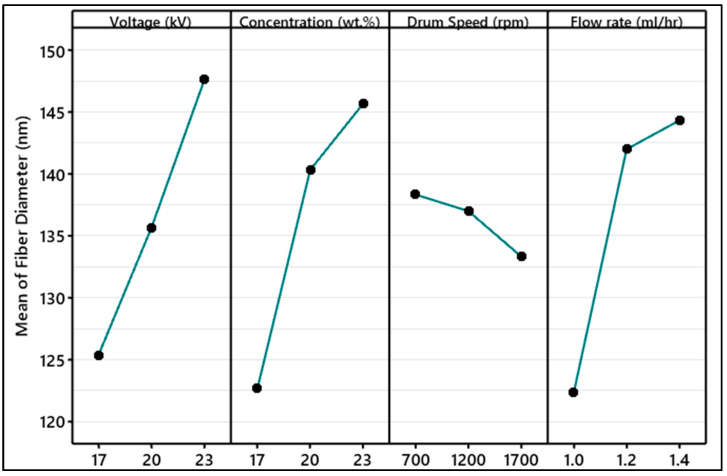
Main effect plots of electrospinning factors on the means of nanofiber diameter analyzed using Taguchi design of experiment.

**Figure 7 polymers-15-04344-f007:**
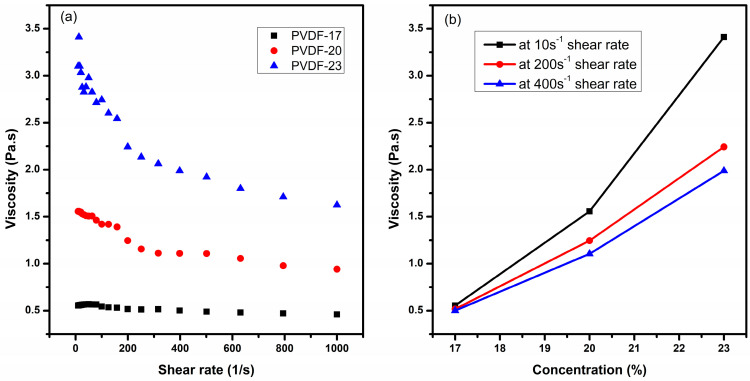
The viscosity of the electrospinning solutions: (**a**) measured in a range of 10–1000 s^−1^ shear rate as a function of PVDF concentration and (**b**) measured at 10 s^−1^, 200 s^−1^, and 400 s^−1^.

**Figure 8 polymers-15-04344-f008:**
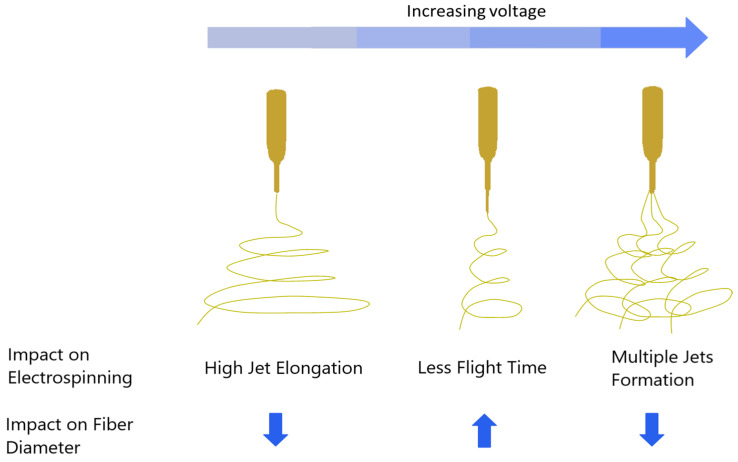
The general effect of voltage on fiber diameter in electrospinning [38].

**Figure 9 polymers-15-04344-f009:**
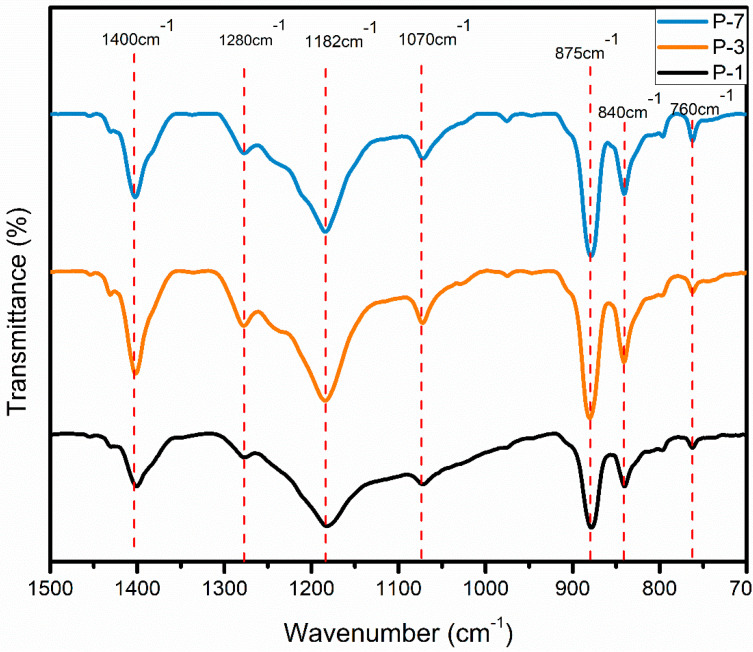
Comparison of FTIR spectra for P−1, P−3, and P−7.

**Figure 10 polymers-15-04344-f010:**
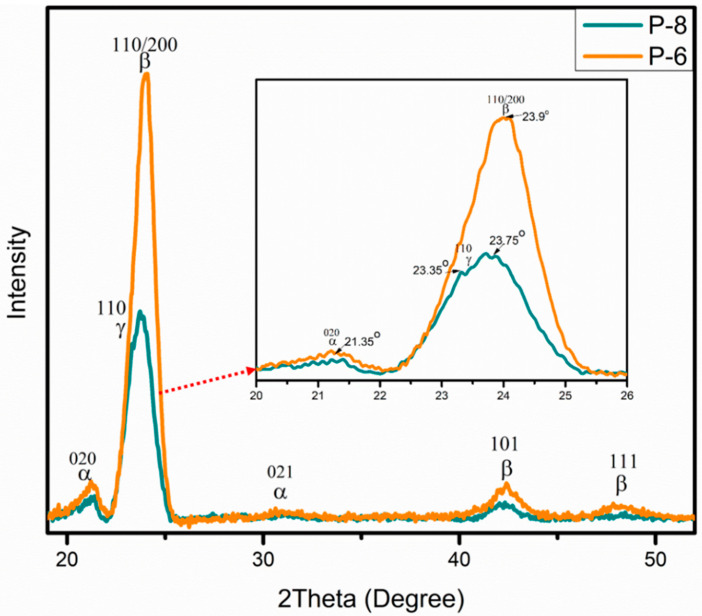
Comparison of X-ray diffraction analysis of P-6 and P-8.

**Figure 11 polymers-15-04344-f011:**
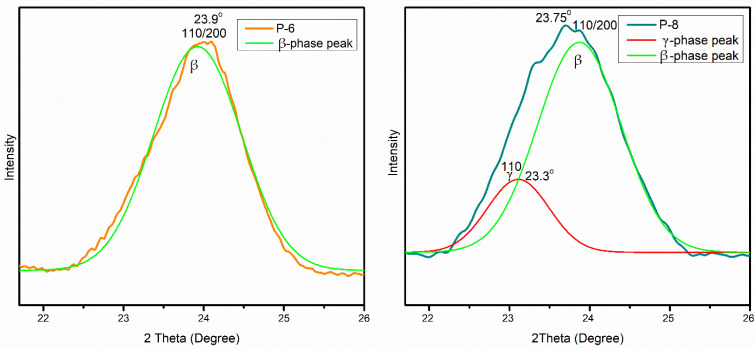
Deconvolution of X-ray diffraction (XRD) of P-6 and P-8 for β-phase and γ-phase.

**Figure 12 polymers-15-04344-f012:**
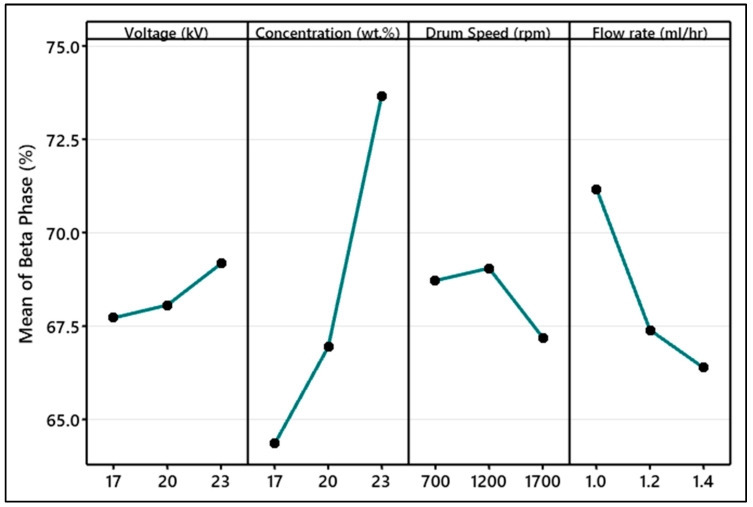
Taguchi analysis of β-phase content (%) affected by electrospinning parameters.

**Figure 13 polymers-15-04344-f013:**
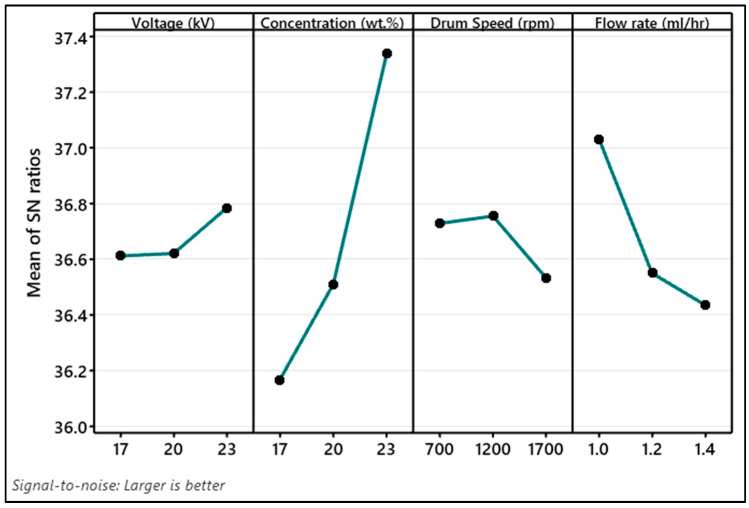
Signal-to-noise ratio (larger the better) graph of β-phase content (%) for electrospinning factors.

**Figure 14 polymers-15-04344-f014:**
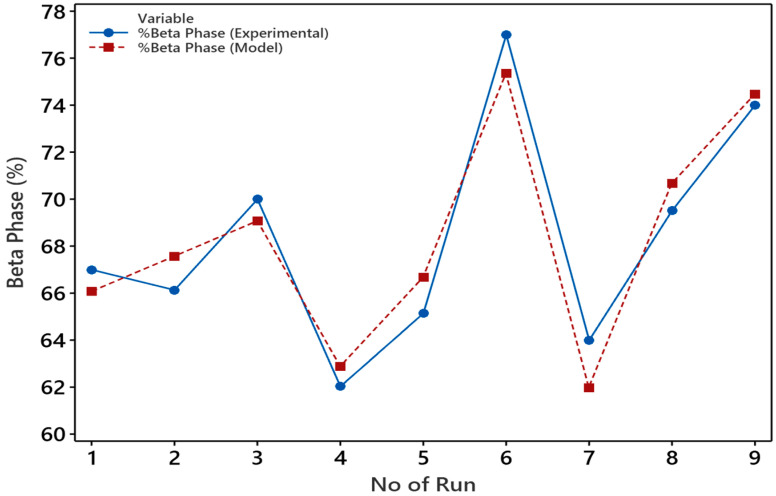
Mathematical model validation with experimental results for β-phase content.

**Figure 15 polymers-15-04344-f015:**
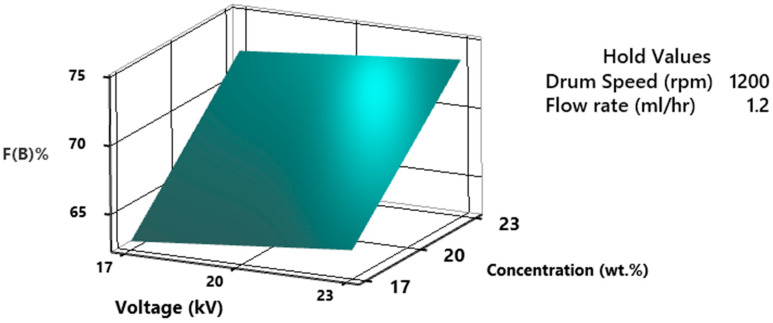
3D surface plot of %fraction of β-phase as a function of concentration and voltage.

**Figure 16 polymers-15-04344-f016:**
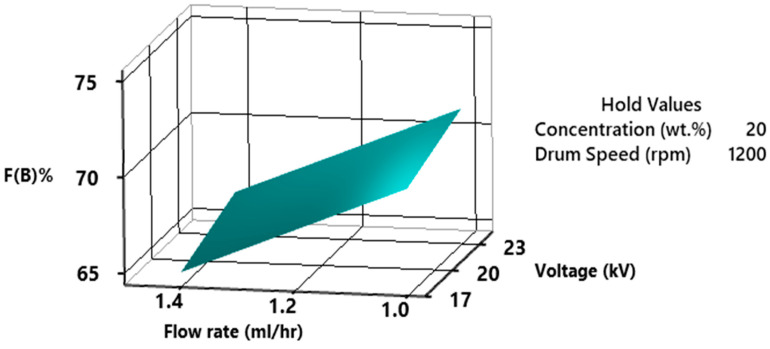
3D surface plot of %fraction of β-phase as a function of flowrate and voltage.

**Figure 17 polymers-15-04344-f017:**
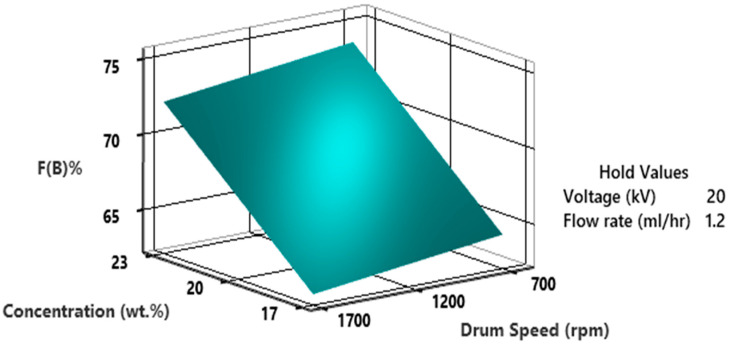
3D surface plot of percentage fraction of β-phase as a function of concentration and drum speed.

**Figure 18 polymers-15-04344-f018:**
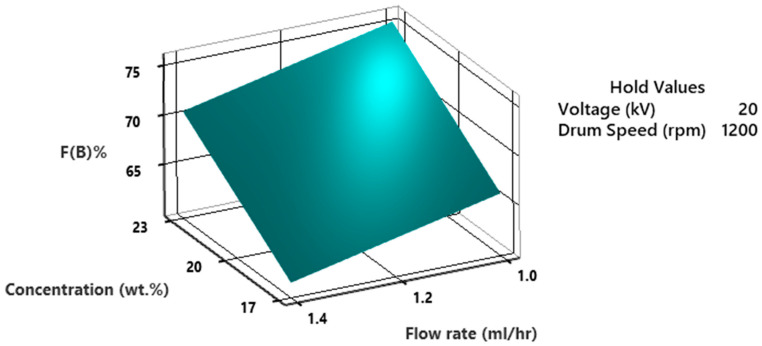
3D surface plot of %F(β) as a function of concentration and flow rate of the polymer solution.

**Table 1 polymers-15-04344-t001:** Electrospinning control factors and levels for PVDF Nanofibers.

		Unit	Level 1	Level 2	Level 3
**Control Factors**	Voltage	kV	17	20	23
	Polymer Concentration	wt.%	17	20	23
	Rotational Speed of Drum	RPM	700	1200	1700
	Solution Flow Rate	mL/h	1	1.2	1.4

**Table 2 polymers-15-04344-t002:** Experimental plan for the highest β-phase content and fiber diameter analysis of PVDF.

Sample Name	Voltage (kV)	Concentration (wt.%)	Speed of the Collector Drum (rpm)	Flow Rate (mL/h)
P-1	17	17	700	1.0
P-2	17	20	1200	1.2
P-3	17	23	1700	1.4
P-4	20	17	1700	1.2
P-5	20	20	700	1.4
P-6	20	23	1200	1.0
P-7	23	17	1200	1.4
P-8	23	20	1700	1.0
P-9	23	23	700	1.2

**Table 3 polymers-15-04344-t003:** Electrical Conductivities of PVDF solutions in DMF.

Sample	Solution Concentration	Electrical Conductivity mS/cm
1	17 wt.% PVDF	1.723
2	20 wt.% PVDF	1.531
3	23 wt.% PVDF	1.165

**Table 4 polymers-15-04344-t004:** Description of various peaks in PVDF.

Frequency	Assignment
1400 cm^−1^	C-H bending due to methylene in aliphatic
1390 cm^−1^	C-CH_2_ rocking
1280 cm^−1^	In the β-phase, C-F_2_ out-of-plane deformation
1230 cm^−1^	In the γ-phase, C-F_2_ stretching
1182, 1172 cm^−1^	C-H rocking
1090, 1070 cm^−1^	C-H in the plane and out of plane deformation
875 cm^−1^	Amorphous phase in PVDF (C-H)
840 cm^−1^	In the β-phase, C-H_2_ rocking
833 cm^−1^	Shoulder, due to γ-phase
760 and/or 660 cm^−1^	In the α-phase, due to C-F_2_

**Table 5 polymers-15-04344-t005:** Comparison of obtained fiber diameter and β-phase of PVDF in the current study with the data in the literature.

Electrospinning Parameters	Fiber Diameter	β-Phase Fraction	Ref.
17–23 kV voltage, 1–1.4 mL/h flowrate, 17–23 wt.% of PVDF in DMF solvent, 700 rpm to 1700 rpm drum speed, 13 cm Needle tip to collector drum distance, 0.26 mmOD needle size.	120–150 nm	65–75%	This work
12–30 kV, 20 wt% PVDF in 1:1 ratio DMAC:Acetone solvent mixture, 0.5 mL/h, flat plate collector at a distance of 20 cm.	151–295 nm Increasing the voltage causes a reduction in fiber diameter.	65–72% Increasing the voltage causes an increase in β-phase.	[19]
14–18 kV voltage, flow rate in a range of 0.5–3 mL/h, 10 wt.% PVDF in 4/6 DMF:Acetone solvent mixture. 0.3–1 needle OD, Collector drum at a distance of 15 cm and 300 rpm.	Not studied	70–78%	[20]
16–26 wt.% PVDF in DMF/acetone (*v*/*v* 4/6), 1 mL/h flow rate, 9–21 kV voltage, Drum speed = 100 rpm, 9–21 cm needle to collector distance.	0.3–0.8 μm	75–85%	[23]
20 wt.% PVDF in DMF/Acetone (1:1, 1:3), 13 and 21 kV voltage, 0.5 and 1 mL/h flow rate, needle to collector distance = 17 cm, 0.75 mm ID of the needle.	300–400 nm	0.6–0.8 fraction	[48]
20 wt.% PVDF in DMF, Varying voltage from 15–30 kV, needle Id 250 μm at 4 mL/h, collector drum at 15 cm distance from needle.	403–495 nm High fiber diameter at low voltage and high flow rate.	Increasing the voltage causes the reduction in β-phase from 85–80%.	[49]
12 wt.% PVDF in DMF:Acetone (100:0, 80:20, 60:40), 0.43 ID of the needle, 250 rpm collector drum placed at a distance of 10 and 16 cm from the needle tip, voltage varied in a range of 9–14 kV, flow rate varied between 0.7 to 0.5 mL/h	495 nm	80% β-phase when electrospun with pure DMF, an increase in the voltage results in an increase in β-phase from 67–72% and increasing the TCD results in higher β-phase. The trend in flow rate was not readily evident.	[24]
Concetration varied from 16–26 wt.% in DMF:Acetone mixture at 4/6 *v*/*v*, 1 mL/h, drum speed of 100 rpm, applied voltage 15 kV; spinning distance 15 cm; nanofiber mat thickness 70 mm	0.2–0.8 μm Increasing the concentration increased fiber diameter.	Increasing the concentration to 20 wt.% gives a higher β-phase (85%) and further increases concentration to 26 wt.% resulted in a lower β-phase fraction (80%).	[23]

## Data Availability

The data used to support the findings of this study are included within the article.

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
