# Peer review of "Statistical Modeling and Optimization of Electrospinning for Improved Morphology and Enhanced β-Phase in Polyvinylidene Fluoride Nanofibers"

_polymers, 2023, doi:10.3390/polym15224344_

Round 1

Reviewer 1 Report

Comments and Suggestions for Authors

Dear Authors,

in litterature many studies can be found of the electrospinning of PVDF. The novelty claimed by this paper is the use DOE to explain the impact of of four factors on three response variable (beta phase, fiber diameter as well as uniformity of fibers).

I recommend to give make a comparison of DOE results found in litterature and those given in this paper. Otherwise the results presented are classical.For example the results shown in figure 6 are well known.

Some remarks and comments:

How have you determined the levels for the differents factors? Have you performed preliminary studies? 

You decided to modify three process parameters and one solution parameter. But this modification implied a modification in terms of viscosity, electrical conductivity, etc. which have an impact of the nanofiber mats produced. It will be interested to give these informations.

Concerning the diameter measurements: have you performed manual measurements or did you used ImageJ plugins?

How on the Fig.6 (or 10) can you conclude on the ranking of the different parameters. By reading this figure it is not clear for me.

Have you performed deconvolution of XRD spectra to separate the beta and gamma phase (2nd peak around 23°)

How can you sustain the hypothesis given in page 10 about the vibrations and air disturbance.

You proposed a mathematical model for the beta response. This model takes into account three process parameters and one "material" parameter. What about the physical sens of such model?

Reviewer 2 Report

Comments and Suggestions for Authors

The manuscript titled "Statistical Modeling and Optimization of Electrospinning for Improved Morphology and Enhanced β-phase in PVDF Nano-fibers" presents a detailed investigation of the electrospinning process parameters to optimize the production of PVDF nanofibers with a high fraction of β-phase.

1.     While the manuscript is useful, there are certain places where the information could be presented more clearly. A few lengthy lines need to be broken down into smaller ones for better understanding. Furthermore, captions for images 4 and 6 need to be elaborated to enhance comprehension.

2.     The paper contains a substantial quantity of material, especially in the form of figures. A more extensive interpretation of the results, highlighting the consequences of various findings and the significance of specific data points, would be good.

3.     The document acknowledges the creation of a mathematical model but doesn't go into detail on the model's creation, accuracy, or propensity for prediction. More information on the model and its validation would boost the research's credibility.

4.     While the article examines the impacts of several factors on the -phase content, a more comprehensive discussion of the real-world implications of the findings would be beneficial. Such as: What are the implications of these adjusted parameters for specific applications in sensors and energy harvesting devices?

In summary, the authors are recommended to focus on clarity of presentation, in-depth data analysis, and a more extensive explanation of the practical implications of the findings to enhance the work. It is also suggested to address language and grammatical difficulties.

Comments on the Quality of English Language

The text has a few grammatical mistakes and poorly written lines. A comprehensive proofreading and editing procedure to resolve these errors will improve the paper's overall quality.

Reviewer 3 Report

Comments and Suggestions for Authors

Manuscript ID: polymers-2675691

Title: Statistical Modeling and Optimization of Electrospinning for Improved Morphology and Enhanced β-phase in PVDF Nano-fibers

The article by Tariq et al. is focused on optimizing electrospinning parameters for piezoelectric PVDF nanofiber membranes.

The manuscript is well-written and quite detailed, and a good contribution to Polymers - MDPI.

Nevertheless, the following remarks and questions should be addressed.

-        In the Taguchi design technique, calculating the signal-to-noise (S/N) ratio is a crucial step. Taguchi describes the S/N ratio as a method for reducing variance and as a performance criterion in experiment design. In the literature several articles report the signal-to-noise (S/N) ratio for the Taguchi method. Why didn't the authors report the S/N ratio in the current manuscript?

 In light of the above considerations, MINOR revisions are suggested to integrate the manuscript with further useful details and increase the added value of the contribution.

Comments on the Quality of English Language

Minor editing of English language required

Round 2

Reviewer 2 Report

Comments and Suggestions for Authors

The authors addressed all the queries raised and this manuscript can now be published.